# Frequency distribution of journalistic attention for scientific studies and scientific sources: An input–output analysis

**Markus Lehmkuhl***, **Nikolai Promies**

Department of Science Communication, Institute of Technology Futures, Karlsruhe Institute of Technology, Karlsruhe, Germany

* markus.lehmkuhl@kit.edu

**Data Availability Statement:** All relevant data are within the manuscript and its Supporting Information files.

**Funding:** This work was supported by grant 411038189 from the Deutsche

## Abstract

Based on the decision-theoretical conditions underlying the selection of events for news coverage in science journalism, this article uses a novel input-output analysis to investigate which of the more than eight million scientific study results published between August 2014 and July 2018 have been selected by global journalism to a relevant degree. We are interested in two different structures in the media coverage of scientific results. Firstly, the structure of sources that journalists use, i.e. scientific journals, and secondly, the congruence of the journalistic selection of single results. Previous research suggests that the selection of sources and results follows a certain heavy-tailed distribution, a power law. Mathematically, this distribution can be described with a function of the form $C*x^{-\alpha}$. We argue that the exponent of such power law distributions can potentially be an indicator to describe selectivity in journalism on a high aggregation level. In our input-output analysis, we look for such patterns in the coverage of all scientific results published in the database Scopus over four years. To get an estimate of the coverage of these results, we use data from the altmetrics provider Altmetric, more precisely their Mainstream-Media-Score (MSM-Score). Based on exploratory analyses, we define papers with a score of 50 or above as *Social Impact Papers* (SIPs). Over our study period, we identified 5,833 SIPs published in 1,236 journals. For both the distribution of the source selection and the distribution of the selection of single results, an exponentially truncated power law is a better fit than the power law, mostly because we find a steeper decline in the tail of the distributions.

## Introduction

This study is about "external effects" which result from the selection of scientific news by individual journalists worldwide. We use this term in the sense of the "model of sociological explanation" ("Modell der soziologischen Erklärung") by German sociologist Hartmut Esser [1]. According to this model, an "external effect" is an unintended result of decisions by a multitude of individual actors, which becomes visible as a social structure by way of aggregation. Our interest lies in the structures of how scientific journals influence journalism and in the structure of the social dissemination of research results.

Forschungsgemeinschaft (DFG, https://www.dfg.
de/en/index.jsp) and the French National Research
Agency (ANR, https://anr.fr/en/). The funders had
no role in study design, data collection and
analysis, decision to publish, or preparation of the
manuscript.

**Competing interests:** The authors have declared
that no competing interests exist.

Methodologically, we will try to uncover these structures in a novel input-output analysis.
The approach is novel mainly due to its scale. Over a period of 48 months between August
2014 and July 2018, a considerable share of all events of the type "new study result" have been
included in the analysis (more than eight million results). The output includes all of these eight
million studies on which journalists around the world have reported to a notable degree. In
our study, we make use of favourable circumstances: on the input side, the publication of
study results and their digital archiving allow for an at least approximative estimate of the
absolute number of all events from which journalism can choose. We selected the database
Scopus as our reference point for this estimation. Scopus currently records about three million
new articles annually [2].

On the output side, new providers of so-called "altmetrics" offer instruments to estimate
the actual selection practice within journalism. Altmetrics are alternative metrics intended to
represent the social impact of scientific results using various indicators, mostly based on usage
data from social networks or other online platforms [3].These indicators can be retrieved by
means of automated data queries, which makes it possible to conduct surveys of selection prac-
tices in thousands of media titles. This is especially valuable for studies like this one, in which
we try to uncover structures created by the selection decisions of thousands of individual jour-
nalists, each of whom aims to attract attention.

Our theoretical concept of journalists' selections of concrete study results is that of a com-
plex individual decision problem [4]. The complexity arises from the interaction of factual,
temporal, and social restrictions which individual journalists have to reduce in order to be
functional [5]. The fact that many journalistic actors worldwide face similar individual deci-
sion-making problems (here: on which studies do I report?) and are basically subject to similar
restrictions must create a structure, a dynamic social order [6]. Before we unfold detailed
hypotheses on this structure, we will describe the factual, temporal, and social restrictions that
affect all journalists at the micro-level, albeit certainly to varying degrees.

## News selection as a complex decision-making problem

From a *factual* point of view, the rational selection of study results is complex for three reasons,
in particular: Firstly, a journalist making a selection does not know the actual number of
options from which they can choose. Secondly, the social relevance of a single research result
is usually likely to be very small and, if there is any, it is usually very difficult for journalists to
assess. Finally, journalists can no longer unconditionally trust the integrity of scientific knowl-
edge production.

According to bibliometric analyses, the number of scientific publications has grown at an
annual rate of about three percent; in some areas, such as open access publishing, the body of
literature has even increased by about 30 percent annually since 2000 [7,8]. Currently, the cor-
pus of scientific articles and reviews in the Elsevier-operated database Scopus is growing by
about 5,500 texts daily. Considering the sheer number of new results, journalists' news selec-
tion cannot possibly be based on the knowledge of all, or even only a considerable part, of this
research output, especially since new study results are only one type of event from which jour-
nalists select [9–12].

The lack of social relevance of any single study finding, or respectively, outsiders' inability
to assess such relevance, might be a direct consequence of the excessive specialisation of scien-
tific knowledge production [13,14]. The social significance of a single result can usually only
be assessed in the light of the results of many other study results, which massively exceed the
processing capacity of news journalism. Alternatively, the significance of a set of research

results can result from assumptions about their future significance: findings x or y might lead to z in the future.

Over the last few years, trust in the integrity of scientific knowledge production has eroded [12,15]. The huge number of scientific publications creates great competition for attention and thus, for publication in a fairly small number of leading journals. Due to their comparatively high impact, these journals greatly increase the likelihood of attracting attention within the scientific community itself. This in turn creates incentives for scientists to resort to unethical means to generate positive research results, as they improve the chances of getting published in high-impact journals [16–20].

Taken together, on the micro level, a science journalist has to make selections from an unmanageable abundance of new scientific results of largely unknown public relevance, which they can only gauge in exceptional, very special cases. Moreover, journalists usually do not have the necessary skillset to assess their accuracy [12].

In terms of *time*, journalists are under great pressure to make decisions quickly and immediately, which increases complexity. Journalistic news selection has always been characterized by considerable time restrictions; (too) much has to be done by (too) few within (too) short a time. We assume that time pressure has only increased over the years, given the established providers' extensive loss of reach, the partial, yet sustained loss of advertising revenues, and the overall economical precariousness of online and offline media [6,21,22]. It is therefore not surprising that journalists from 16 countries all over the world subjectively emphasize the significance of "procedural influences" on their actions, which Hanitzsch summarizes as "time scarcity", "standards and procedures of news production", as well as "lack of resources" [23].

In science journalism, too, processes have been accelerating, catalysed not only by changes in economic parameters, but also by technical advances in the distribution of scientific results [24]. This is supported by five qualitative surveys of science and health journalists that asked about "constraints". Shortage of time turned out to be the most frequently mentioned factor limiting the scope for action [25]. Thus, time restrictions also have a profound impact on the actions of science journalists.

Lastly, *social* restrictions arise from the influence of third parties on a journalist's individual decision-making. Three main interaction contexts generate complexity: Interactions with other journalists, interactions with sources, and interactions with recipients.

**Interactions with other journalists.** The selection of an event by an editor (and thus, the non-selection of another) must prove itself in the light of the decisions made by all other editors (that are considered to be relevant) no later than the following day. The fact that journalists have to ask themselves whether other journalists will make the same selection decision increases complexity [26]. On the one hand, journalists fear that they might "miss" events. On the other, the choices made by competing media can also lead to a negative selection decision when journalists strive for exclusivity [27].

Unfortunately, very few studies support a reliable assessment of the importance of peer orientation in science journalism. Taken together, individual observational studies, surveys, and content analyses [24,27–33] indicate that science journalists also carefully observe their colleagues' work. In surveys, leading international titles like the New Scientist, the BBC, or the New York Times are even mentioned as important sources for the selection of news topics, alongside renowned scientific journals such as Nature and Science. However, we lack any evidence that science journalists' selection of study results is significantly influenced by what they believe their colleagues are selecting. The findings suggest that science editors tend to be unaffected by their competitor's choices when selecting topics for their own science coverage. This in turn can be partly explained by the fact that the science system rarely produces study results with significant news value [13].

**Interactions with sources.** It can be assumed that individual science journalists, at least in Europe and the US, are confronted with increasingly professional science PR departments that try to influence journalists' selections by adapting to their routines [34,35]. This is probably mainly due to a political process commonly termed "new governance of science" [36]. This term refers to the state partially withdrawing from micro-governance of science in favour of a project-based control system aimed at specific objectives. On the one hand, this new form of governance gives science organisations greater freedom, but on the other, puts them in a more competitive situation. Politics has geared science towards greater and more international competition, which is generally considered an engine of the medialization of science. In order to drive home the social relevance of one's own actions to political elites, coverage, especially in elite media, becomes more important for scientists and their institutions [37,38]. This assumption is supported by further study results [38–47]. Surveys, analyses of PR activity, qualitative case studies, and input-output analyses unanimously point to the fact that a "primacy of self-promotion" [48] seems to be guiding the actions of scientific organizations and scientific publishers.

**Interactions with recipients.** Among the social groups mentioned so far, which presumably influence science journalists' topic selection, recipients are certainly the most important group. The coordination of selection actions by journalists and recipients has become considerably more complex [49–51]. Journalism has suffered a noticeable loss of control over what its audience receives. This process is catalysed by a proliferation of non-journalistic offers and the technologically enabled transformation of a formerly passive audience into an active producer of public messages [52–54]. The degree of this loss of journalistic control seems to vary from country to country. It seems to be particularly large in the USA, where only about one fifth of respondents in representative surveys state that they refer to journalistic sources for information on science and technology [55,56], while in Germany, for example, a slight majority still mentions journalistic sources [52,57].

Such numbers illustrate the extent to which the bond between journalism and a sufficiently large audience is eroding. Accordingly, in addition to event-, topic-, and product-related characteristics, user and usage characteristics are increasingly considered guiding factors for journalistic selections, especially in online journalism [58,59]. The need to let recipients participate directly in topic selection [60] makes the selection process even more complex. There are some indications that topic selection in science journalism is strongly influenced by public preferences on the micro level. While we must assume that topic selection in politics, at least in the Western world, is determined by normative claims of parliamentary democracy, this applies at best conditionally to the current "result journalism" on scientific topics. In addition, the relatively high appeal of scientific contents for audience segments of higher education and income levels might lead journalists to strongly align their selections with these groups' preferences [61].

In summary, the current state of research does not permit more comprehensive insights into the coordination of news selection between science journalism and recipients [52]. But we can certainly assume that in order to protect the bond with its audiences, journalistic selection requires more mechanisms than a mere orientation towards news value [62].

So far, we have made cursory efforts to outline science journalists' individual decision-making situations. According to the model of sociological explanation [1], this insight is necessary to explain the structures resulting from the collective actions of journalists as they select research results. The following section provides a detailed overview of the state of research on these structures.

**Table 1. Content analyses that have examined science journalists' source choices.**

| Most important journals, ranked by the number of references found in the respective analysis | | | | BMJ | | Nature | | |
|---|---|---|---|---|---|---|---|---|
| | | | | Nature Medicine | | Journ. of Clinical Oncology | | |
| | | | JAMA | JAMA | | PNAS | | Lancet |
| | | | Nature | PNAS | | Cancer | Neurology | Archives of Internal Medicine |
| | | | Science | New Scientist[a] | | Lancet | Circulation | JAMA |
| | BMJ | JAMA | BMJ | Lancet | Cell | Science | Nature | NEJM |
| | Nature | NEJM | Lancet | NEJM | Nature Genetics | JAMA | Science | PNAS |
| | Lancet | Nature | New Scientist[a] | Science | Nature | Journ. of the National Cancer Institute | JAMA | Science |
| | NEJM | Science | NEJM | Nature | Science | NEJM | NEJM | Nature |
| **Study** | van Trigt et al. 1994 [11] | Evans 1995 [66] | Semir 1996 [67], Semir et al. 1998 [68] | Pahl 1997 [33] | Bubela & Caulfield 2004 [69] | Moriarty, Jensen, & Stryker 2010 [10] | Suleski & Ibaraki 2010 [70] | Kiernan 2016 [41] |

[a] While the New Scientist was identified as one of the most important external sources for the journalistic coverage of biomedical research in [67], it differs from the other sources listed in the table in that it does not publish original research. This also explains why the New Scientist cannot be found in the results of our analyses.

## The structure of news selection

As we have mentioned above, we are interested in external effects arising from individual selection decisions by a large number of science journalists who are all subject to similar restrictions. Specifically, we are interested in the structure of source selection and the structure of dissemination of scientific results.

With regard to *sources*, we can assume, supported mostly by content analyses, but also based on surveys [9,63–65], that the above-mentioned restrictions on individual selection decisions have led to a clearly pronounced preference for certain journals. In Table 1, we have summarised all content analyses that offer valid information about journalists' preferred sources.

The selection is primarily based on the journals Nature, Science, New England Journal of Medicine (NEJM), Lancet, and the Journal of the American Medical Association (JAMA), albeit in significantly different orders, depending on the sample of media titles analysed. We can assume that science journalism clearly prefers these scientific journals, even though so far, examinations have focused mostly on the selection of medical study results. A large part of journalistic news selection seems to draw on only a small number of scientific journals, which in turn publish only a very small number of the studies submitted to them, while the vast majority of submissions is rejected without review [71].

Four of the analyses also contain quantifications [33,41,68,70], which–unnoticed by the authors–indicate a very specific frequency distribution of journalistic source selection. Semir [68] found that one third of all study results selected by seven large international media titles came from only four journals. Suleski and Ibaraki [70] and Kiernan [41] also noted the dominance of just a few journals. In addition, these two studies suggest that this dominance has weakened over time in favour of references to lesser-known journals. Finally, in her analysis of medical reporting by eight German media titles, Pahl [33] found that three-quarters of the total of 408 references to scientific journals cite only ten journals, about half of the references

only three: Nature, Science, and the New England Journal of Medicine (NEJM). This means that the bulk of journalistic attention was concentrated on a very small number of about eight to ten journals, while all other sources were considered only sporadically in selection decisions. Based on the work of Pahl [33] and Kiernan [41], the frequency distribution of journalistic source selection can be inferred, but only approximately, because the results are not fully documented. In both cases, it is a curve that slopes linearly on logarithmic axes from rank to rank and shows a characteristic, heavy-tailed form (Fig 1, dotted trend lines).

Similar left-skewed distributions have been observed in a multitude of other phenomena, mostly unrelated to journalism, and can be reproduced so reliably that the term 'distribution laws' has become established to describe them. These include, for example, use of websites, the number of citations attributable to scientific articles, or the productivity of scientists. Some of these relationships have been known for a long time, such as Zipf's law, the Pareto distribution, Lotka's law, and the power law [72]. These distributions, although differing in detail, describe the very regular exponential decrease of a quantity y in relation to the linear increase of the logarithm of a quantity x. For example, Lotka's law of scientific productivity states that for every 100 authors of a single scientific paper in a subject area, there are $100/2^2$, i.e. 25, authors who publish two papers, $100/3^2$, i.e. about 11, who publish three, etc. The number of authors decreases by approximately $1/x^2$ with each additional scientific paper [73,74]. A characteristic feature of these power law distributions is that they form a straight line when plotted on logarithmic scales [72].

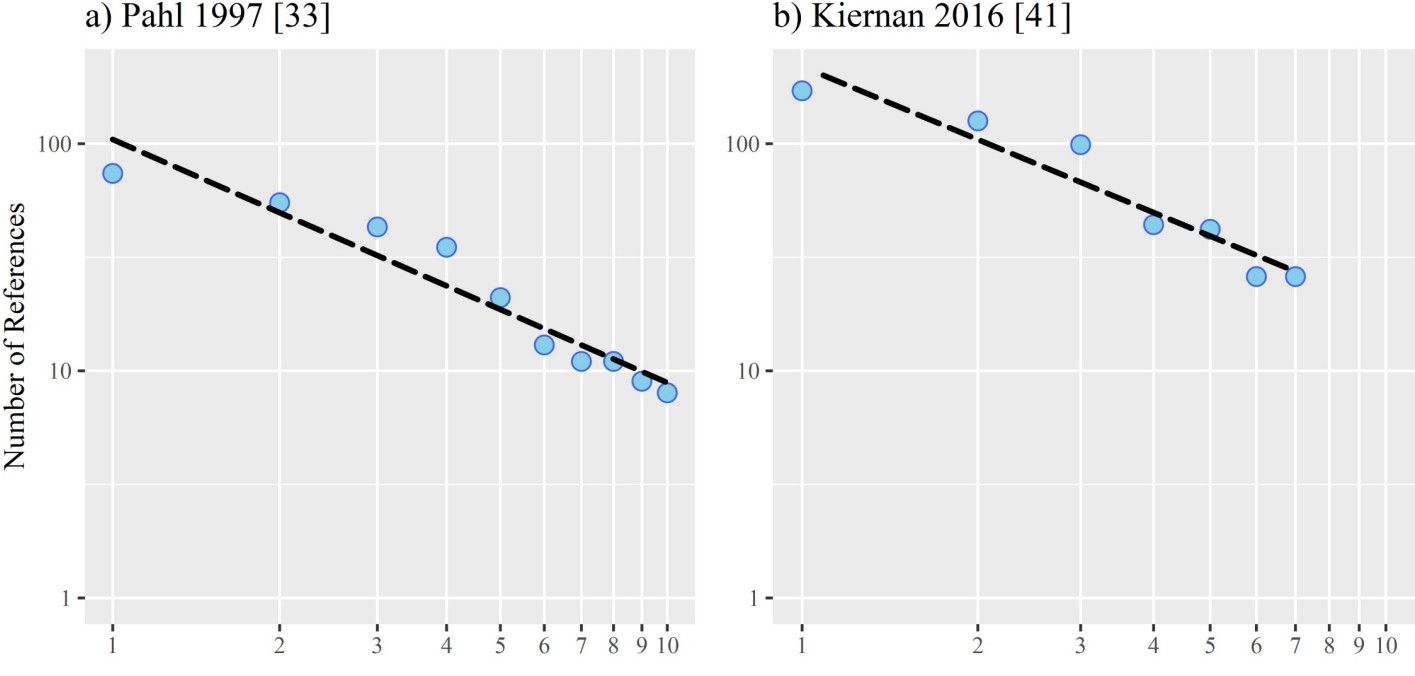

**Fig 1. Frequency distribution of journalistic references to scientific journals by a) eight German science beats between 1995 and 1996 (N = 408) and b) the New York Times (N = 545) between 1998 and 2012 [33,41].** Source: Own presentation based on data a) from [33]: 408 references were analysed; these were for 1. Nature (74), 2. Science (55), 3. NEJM (43), 4. Lancet (35), 5. New Scientist (21), 6. PNAS (13), 7. JAMA (11), 8. Nature Medicine (11), 9. BMJ (9), 10. Münchner Medizinische Wochenschrift (MMW) (8), and 128 other journals (number of references missing in the publication) and from b) [41]: 1,054 references were analysed; these were for Nature (171), Science (126), PNAS (99), NEJM (44), JAMA (42), Archives of Internal Medicine (26), Lancet (26), and 267 other journals (number of references missing in the publication).

In the case of journalistic references to scientific journals, the exponent of the curve is close to one. In both studies, the distribution of references to journals by German and US journalists appears to have the same structure. If the top journal achieves 100 references, then the next journal achieves approximately $100/2^1$, i.e. 50 references, the third $100/3^1$, i.e. 33 references and so on. The data from Suleski and Ibaraki [70] also show approximately the same structure, even though they only collected 121 references to journals in US television programmes and a news magazine. In their study, the number of references decreases by almost $1/x^1$ relative to the number of journals, too.

These findings, although incompletely documented, give rise to the assumption that the distribution of journalists' source choice follows a power law, i.e. that the individual restrictions on journalists as outlined above lead to a structure that can be predicted quite precisely. Accordingly, we formulate hypothesis 1 based on the latter two studies, in particular:

*H1*: The distribution of the influence of scientific journals on journalism, measured by the number of studies that were selected from a given journal, follows a power law distribution.

Unlike the choice of sources, the *structure of the social dissemination* of research findings must be indirectly inferred. As an indicator, we can use the extent to which the same research result is reported by several media titles. A greater congruence in journalists' selection of individual study results indicates a higher degree of social dissemination [75,76].

Relevant science journalism research has so far mainly dealt with the structure of the social dissemination of study results at a higher level of aggregation. For example, it has been repeatedly established that science journalists clearly favour results from the wider realm of medicine, while results from mathematics, for example, are very rarely selected for coverage [50]. The approach pursued here does not start at this meso level but at the micro level. We are interested in the structure of journalistic selection of single study findings, resulting from parallel decision-making processes by individual journalists. To derive concrete hypotheses, we rely on a heterogeneous group of studies from which to derive indications of the structure of journalistic selection since they have determined how many media titles have selected the same event or occasion congruently.

To our knowledge, the most elaborate content analysis on this question was presented by German communication scholar Rössler [77–80]. In four individual studies, he examined the congruency of news selection in daily newspapers and TV news programmes, both nationally and internationally. Beyond that, there are analyses which refer more specifically to lead stories of German daily newspapers [81], respectively to the congruency of the front pages of quality newspapers [82], in addition to another older study [83] from which we can reconstruct the congruency of the selection of national political events. Finally, there are two student theses [32,84] dedicated to congruency in the science departments of five and eleven different media titles respectively.

Although all these works differ in methodical details, they present a similar pattern: The probability that the same event is chosen by one, two, three etc. media titles decreases approximately by $1/x^\alpha$ with each additional media title. The exponents of the trend lines partly differ, depending on the sample and approach used in each analysis. In the science departments, the trend line drops most steeply; in the politics department, it is the flattest. This suggests that policy beats are more likely to pick up the same events than science beats. However, the distribution pattern is the same in all analyses. The structure can best be illustrated with Rössler's work because he has examined the largest sample of media titles (Fig 2).

Although Rössler's results are poorly documented, all four studies show that the proportion of events decreases exponentially relative to the logarithm of the number of media titles that select the same event (i.e. with the degree of congruency). However, there are clear differences in the exponent, depending on the media sample. The exponent of the distribution is much

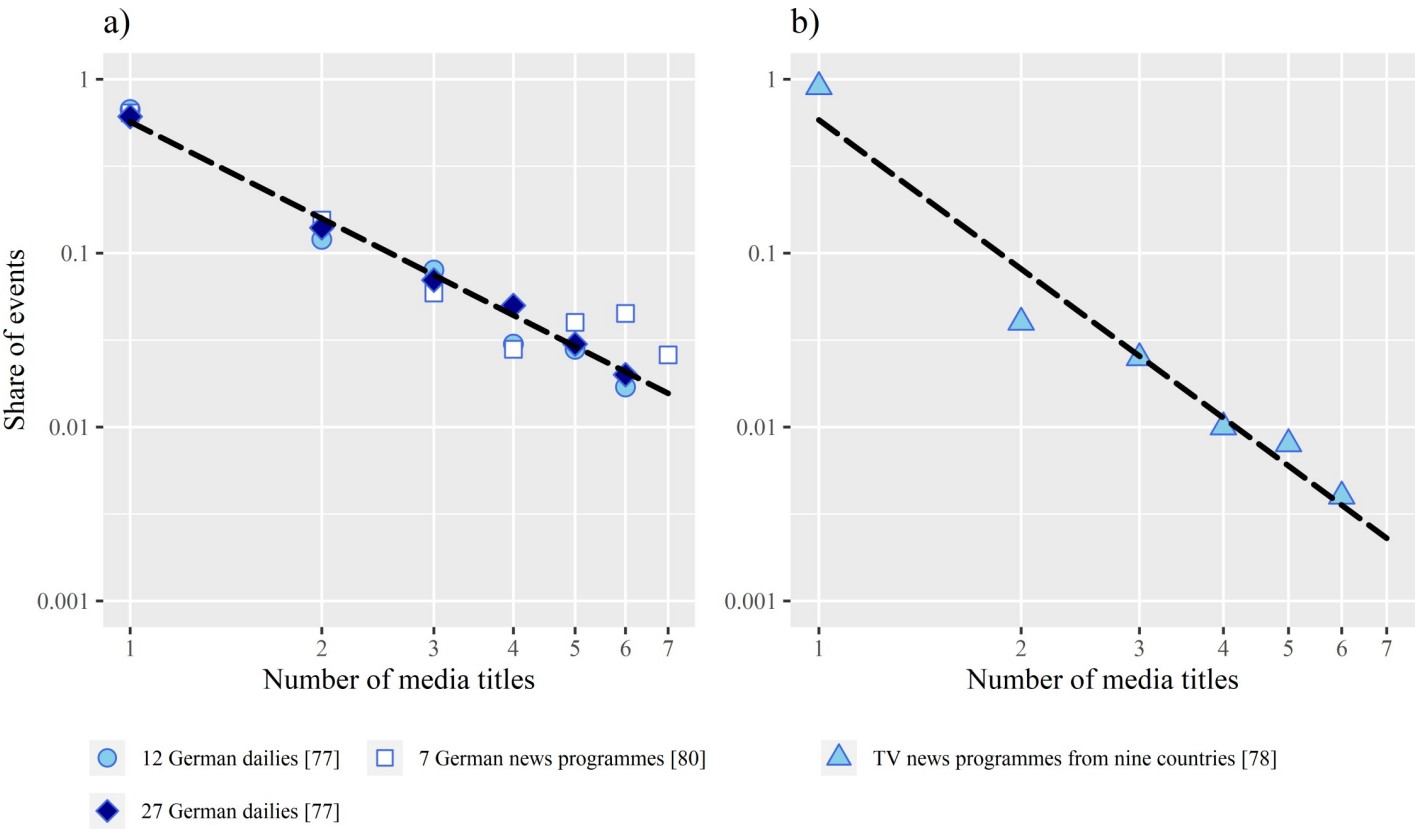

**Fig 2. Distribution of parallel selections of specific occasions in one international and three national studies.** Source: own presentation based on data from [77,78,80]. A) Data from three studies of German newspapers and TV news programmes. B) Data from an international study of TV news programmes. We only show shares for up to seven media titles because in Rössler's studies, shares of events reported by more than seven media titles were only presented in aggregated form.

higher in the international study that compared the selection of topics for TV news programmes in nine countries within a week. Ninety percent of the events were taken up only in a single national context, 4 percent in two, 2.5 percent in three etc. The probability decreases by approximately $1/x^{2.8}$.

Regarding the national studies, it is noteworthy that regardless of the number of media titles, the exclusive share remains approximately the same. What is also surprising is that the proportion of events that are covered by two, three outlets etc., also remains the same. Regardless of the number of media titles, the share of congruent selections falls by approximately $1/x^{1.8}$.

In summary, these findings suggest that the congruent selection of concrete events by x media titles seems to follow a power law or similar heavy-tailed distribution, as well. With every additional media title, the probability of selection seems to decrease by $1/x^{\alpha}$. Accordingly, we formulate:

*H2*: The probability that a study result will be selected congruently by several media titles decreases exponentially with the logarithm of the number of media titles and follows a power law distribution.

## Materials and methods

This study determines which study results published in which journals between August 2014 and July 2018 have been disseminated by journalists to a notable extent. All approximately

eight million articles and reviews listed in the Scopus database in this 48-month period were selected as input. To determine the output, we used data collected by the altmetrics provider Altmetric [85].

We used Altmetric's mainstream media score (MSM). This score indicates in how many of currently over 2,000 online news portals a study was referenced. This is determined by searching for the digital object identifier (DOI) of the study and, for English-language media, for the title and authors of the study. An MSM score of 15 indicates that the study was mentioned in 15 different online portals. The list of online media on which the score is based was compiled by Altmetric. Not every media portal searched by Altmetric indicates journalistic selectivity. The MSM score includes mentions in an unspecified number of aggregator websites and mere disseminators of press releases like EurekAlert. For this reason, small MSM values, in particular, are not suitable indicators for journalistic selectivity.

Therefore, two validations were carried out prior to the actual study to determine whether and at what level the MSM score could be interpreted as an indicator of significant journalistic attention for a study result. The first validation was based on 1,001 study results that were published in the journals Science and Nature between August 2016 and July 2017. The MSM scores of this sample were assigned to 11 groups, scores of 1–9; 10–19; 20–29;. . .; >100. Subsequently, we conducted a manual search of randomly selected sets of five studies per category (N = 55) in the full-text press database Nexis to determine from which score we could infer a resonance on three large national media markets. The media markets selected were the United Kingdom, the United States, and Germany. The response was classified as "noteworthy" if at least ten articles on at least one of the five studies per category had appeared in these media markets. This was the case for scores > = 50. For this reason, we speak of a notable response or "Social Impact Papers" (SIP) when a study has an MSM score of at least 50.

In a second validation, we checked whether studies for which Altmetric does not provide an MSM score have received journalistic attention. This validation was based on nearly 1,000 papers published in the medical field between October and December 2016 for which we did not receive any data in our automated queries. From these papers, a random sample of 60 studies was drawn and manually checked to see if press reports could be found for any of the study results. This search was based on the Nexis database's list of all news sources, which includes thousands of full-text media titles from all over the world. No reports could be found in any of the 60 cases. Based on these findings, we assume that studies for which no data are available did not achieve any relevant journalistic response.

Another issue with the MSM score is that although Altmetric does not publish detailed information, it is safe to assume that the source base has grown steadily since the company was founded in 2011. Even though we assume that the distribution should prove to be robust against changes in the number of media titles, we will compare the number of SIPs and journals with at least one SIP in the different time periods of our analysis in the results chapter to quantify the changes over time.

Based on the validations, the output in our analysis was defined as all studies with an MSM score greater than or equal to 50. The output was determined by automated queries to the API of Altmetric.com. For each study, we retrieved the data collected by Altmetric for this DOI. If the MSM score of a study was above the specified threshold, the study, including the MSM score, was transferred to the output data set. This new data set of all Social Impact Papers from mid-2014 to mid-2018 was analysed to determine the frequency distributions of journalistic selection with respect to the use of sources and to congruent selection of single studies by several media titles. We used several steps to test the hypothesis that a very specific type of heavy-tailed distribution, a power law, is evident in our data. In Figs 1 and 2, we used a least squares fit to get a first estimate of a possible power law form and to generate our hypotheses, a method

which turned out to be unreliable [72,86,87]. For this reason, we used the more accepted methodology presented by Clauset et al. [86] to estimate the parameters and plausibility of a power law distribution in our data.

The first step in this approach is usually to estimate the parameter $x_{min}$, which yields the lower bound of the power law behaviour in the data. To find $x_{min}$, a Kolmogorov-Smirnov (KS) approach is used that calculates the maximum distance between the CCDFs of the data and the fitted model for different values of $x_{min}$. The estimate of $x_{min}$ is the value that minimizes the distance. Because we assumed that the power law behaviour would be present in all of our data, we did not use this approach and fixed $x_{min}$ to be 1 for the distribution of journals and 50 for the distribution of MSM-scores. Once we know from which value we can fit a power law distribution to our data, we can estimate the exponent of the power law distribution $\alpha$ with a maximum likelihood estimator (MLE). To get a measure for the uncertainty of these estimates, we use a bootstrapping approach in which we estimate $x_{min}$ and $\alpha$ for a number of synthetic datasets generated from the original data and take the standard deviation over all datasets [86]. We used 10,000 simulations for this step. The results of the bootstrapping supported our assumption concerning $x_{min}$ because for almost all years our fixed values were within the standard error of the values calculated with the KS-approach.

After these steps, we merely have more reliable estimates of the parameters of a possible power law fit to our data, but we do not yet know whether this fit is plausible. To get a visual impression of the plausibility of the power law fit, we calculated the complementary cumulative distribution function (CCDF) of our data. The CCDF shows the probability that the value of x is at least x, for example the probability that a journal had at least x SIPs. This way of plotting the data is especially useful to identify a potential power law distribution in the data that would show itself as a straight line on logarithmic scales [72]. We created CCDFs for our different datasets and plotted the related power law fits. For a less subjective estimate of the plausibility of the power law fit, Clauset et al. propose two techniques. The first is a goodness-of-fit test, again using the KS statistic which tests the hypothesis that the observed data are drawn from a power-law distribution [86]. We used 10,000 simulations here, as well. The second approach is to check whether other heavy-tailed distributions are a better fit for the data with a likelihood ratio test. The results of this test are the log-likelihood ratio of the two compared distributions (the sign of the ratio shows which distribution is the better fit) and a p-value showing if the sign of the ratio is statistically significant, using Vuong's test (if p is close to zero, the sign is a reliable indicator). Possible alternative distributions with which to compare the power law are an exponential or stretched exponential distribution (Weibull), a lognormal distribution or an exponentially truncated power law, i.e. a power law distribution with an exponential cut off following a function of the form $x^{-\alpha_*}e^{-\lambda x}$.

In our analysis, we used the implementation of these methods in the R [88] package poweRlaw [89] and in the Python package powerlaw [90]. The R package was used to estimate the parameters of the power law distribution and to calculate the goodness-of-fit test, while the Python package was used to compare the power law fit with alternative distributions because it implements more alternatives, most importantly the truncated power law, which is missing in the R package. We used the versions of the methods adjusted for calculations with discrete data.

## Results

Of the almost eight million studies available in Scopus for the study period between August 2014 and July 2018, 5,833 had an MSM-score of 50 or more. Thus, we assume that fewer than one in 1,000 of these new scientific results were reported by international journalism to a

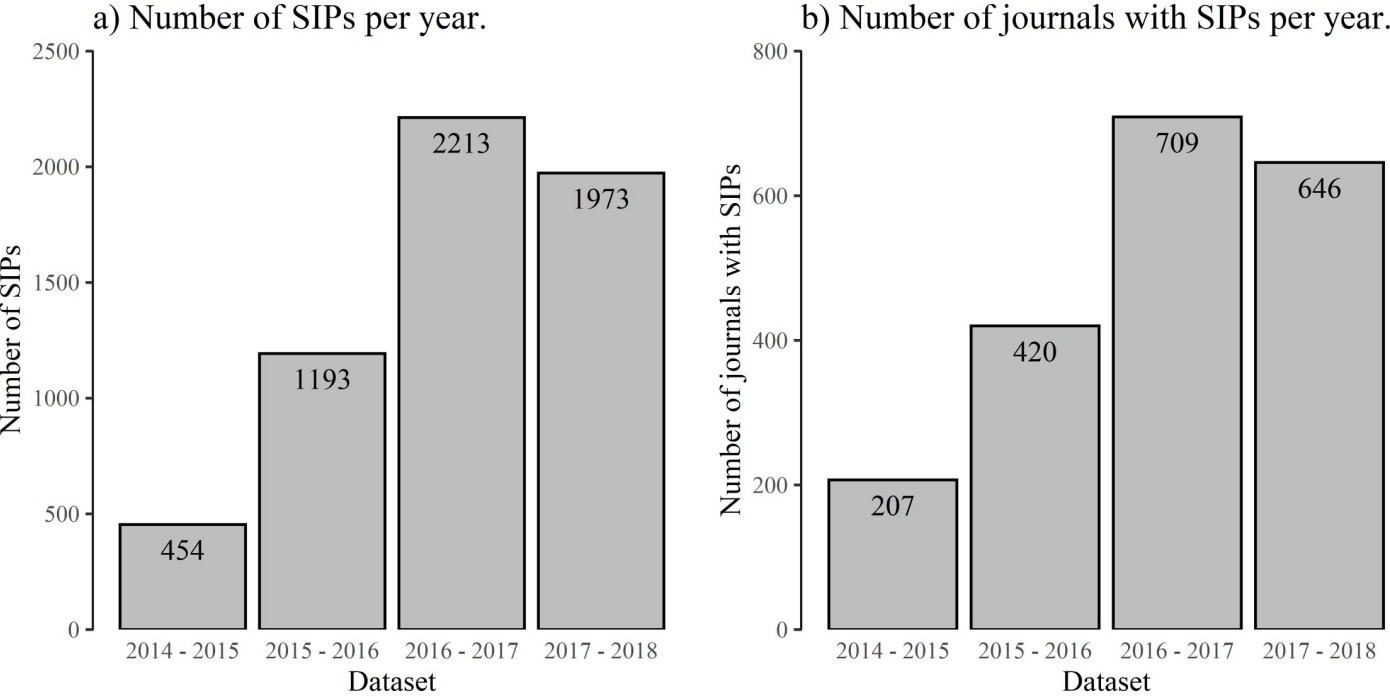

**Fig 3. Development of the number of Social Impact Papers (SIPs) and journals with SIPs.** A) Comparison of the number of studies with MSM score > = 50 in the different years of the study period, showing more than a quadrupling from 2014 to 2016. B) Comparison of the number of journals with at least one SIP in the different years of the study period, with a similar development as the number of SIPs.

notable extent. The number of Social Impact Papers (SIPs) and journals with SIPs has changed over the years in our study period, showing a strong increase from 2014–2015 to 2015–2016 with a doubling of both numbers and again from 2015–2016 to 2016–2017 (Fig 3). We assume that this is due to changes in the source base that Altmetric uses to calculate the MSM score, but since Altmetric does not publish information concerning the development of their sources, we cannot test this assumption. Because of the large yearly differences, it is difficult to compare the results from the different years in our study period. While we will still present the results for each year individually, we will focus on the results of the last two years in our figures and use only these years for the aggregated fits because the data was comparable over this period of time.

Looking at the journals in which the Social Impact Papers (SIPs) were published, we find a very unequal distribution. 1,236 journals published at least one study with an MSM-score of 50 or more. 55 percent of these journals (696) had only one SIP over the four-year period. The ten most influential journals published 27 percent of all SIPs: Nature (274), Science (257), NEJM (203), PNAS (201), PLOS One (141), The Lancet (130), JAMA (122), Nature Communications (115), Pediatrics (113), and Scientific Reports (109).

Using the methods from [86], we estimated a power law fit with the data from the last two years, which resulted in an exponent of $\alpha = 1.93$ with the power law-behaviour starting at $x_{min}$ = 1. According to this estimate, the probability that a journal with n SIPs will publish another SIP decreases regularly by $x^{-1.93}$. Fig 4A and 4B show the distribution of the number of journals with a certain number of SIPs for the last two years of our study period, transformed into a complementary cumulative distribution function. The x-axis shows the number of SIPs in a journal and the y-axis the share of journals with at least this number of SIPs, i.e. all journals in our dataset contained at least one SIP, leading to a share of 1 for x = 1. Only about 1 percent of

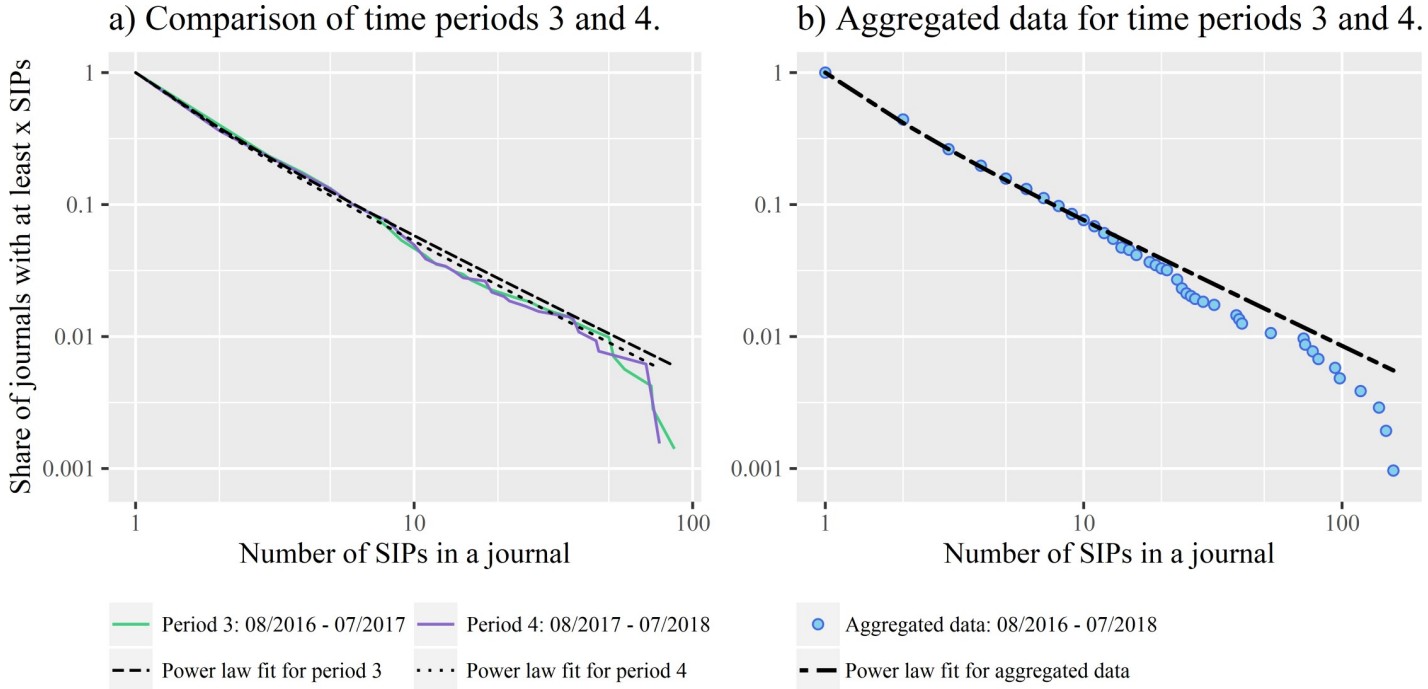

**Fig 4. Complementary Cumulative Distribution Function (CCDF) of Social Impact Papers (N = 4,186) on journals between August 2016 and July 2018 (N = 1,036).** A) CCDFs and power law fits for the share of journals with a certain number of SIPs for the last two years of the study period (N = 709 for the number of journals from August 2016 to July 2017 and N = 646 for August 2017 to July 2018). The distributions are very similar. B) CCDF and power law fit for aggregated data from panel A).

all journals had at least 100 SIPs, thus for x = 100, P(x) = 0.01. Fig 4B shows the aggregated data. The data points very closely follow the dotted line representing the power law fit until there is a drop-off at the last data points.

The result of the goodness-of-fit test is a p-value of 0.19 (Table 3), thus we do not reject the power law hypothesis for the distribution of SIPs on journals. We also compared the power law fit with alternative heavy-tailed distributions to see if there was a more plausible fit. While the power law is a better fit than an exponential distribution, the lognormal and Weibull distribution as well as a truncated power law with an exponent of 1.85 seem to be even more plausible models of the data (Table 2). The results of the log-likelihood tests between the truncated power law and the other two candidate distributions are inconclusive. Given these results, we cannot fully confirm hypothesis 1. While the power law hypothesis was not rejected as result of the goodness-of-fit test, other distributions are better fits.

**Table 2. Results of likelihood ratio tests for different heavy-tailed distributions.**

| Data | Comparison power law–lognormal distribution | Comparison power law–exponential distribution | Comparison power law–Weibull | Comparison power law–exp. trunc. power law | Comparison exp. trunc. power law–lognormal distribution | Comparison exp. trunc. power law—Weibull |
|---|---|---|---|---|---|---|
| SIPs per journal, last two years aggregated | -23.83, < 0.0001* | 498.20, < 0.0001* | -23.76 < 0.0001* | -24.65, < 0.0001* | 0.82, 0.30 | 0.90, 0.21 |
| Number of studies with a specific score, last two years aggregated | -28.50, < 0.0001* | 86.20, < 0.0001* | -30.68, < 0.0001* | -34.43, < 0.0001* | 5.93, < 0.0001* | 3.75, < 0.0001* |

The first value in each cell is the log-likelihood ratio of the two compared distributions, for negative values, the second distribution is the better fit. The second value shows if the sign of the ratio is significant.

**Table 3. Parameters of power law and truncated power law fits and results of goodness-of-fit tests for the distribution of SIPs per journal in the different time periods.**

| Data | Power law: α (SE) | p-value | Exp. trunc. power law: α (SE) | Exp. trunc. power law: λ (SE) |
|---|---|---|---|---|
| SIPs per journal, last two years aggregated | 1.93 (0.074) | 0.19* | 1.85 (0.22) | 0.0066 (0.032) |
| SIPs per journal, 2014–2015 | 2.50 (0.15) | 0.0030 | 2.50 (0.50) | 0.000000014 (0.046) |
| SIPs per journal, 2015–2016 | 2.22 (0.10) | 0.53* | 2.22 (0.32) | 0.00040 (0.045) |
| SIPs per journal, 2016–2017 | 2.04 (0.080) | 0.095 | 1.94 (0.41) | 0.011 (0.057) |
| SIPs per journal, 2017–2018 | 2.07 (0.047) | 0.32* | 2.01 (0.39) | 0.0034 (0.055) |

Standard errors of the parameters were estimated by bootstrapping.

*For p-values above 0.1, we do not reject the power law hypothesis.

The distribution reacts relatively robustly to changes in source numbers. There are no large differences in the parameter of the distribution in the individual years, although the number of journals with SIPs has increased from 207 to 709 since 2014, probably caused by an increase in the number of sources tracked by Altmetric. The exponents of the power law fits have similar values in the individual years, with 2014 to 2015 being the only outliers (see Table 3). For August 2015 to July 2016 and August 2017 to July 2018, the power law is a plausible fit with p = 0.53 and p = 0.32.

In summary, the individual restrictions on journalistic selection of scientific papers lead to a long-tail distribution that comes close to a straight line on logarithmic scales but shows a steeper drop-off in the tail. A power-law distribution with an exponent of about -2 is not an implausible model of the selection of scientific sources, but other distributions are better fits.

Looking at the distribution of congruent journalistic selections, most of the studies have an MSM score close to 50 (median = 66), but the range goes up to values of over 400. Again, we find a skewed, heavy-tailed distribution. The MLE of the exponent for a possible power law fit for data from the last two years of the study period has a value of 3.45 with $x_{min}$ = 50. In the CCDF, the data resembles a straight line on log-log-scales, but there is a steeper drop-off in the tail, leading to a larger deviation from the characteristic power law-form (Fig 5).

This time, the goodness-of-fit test disproves the power law hypothesis and the likelihood ratio tests show that other distributions are a better fit than the power law. While the stretched exponential and lognormal distribution are more plausible than the power law, a truncated power law seems to be the best fit. The truncated power law fit has an exponent of 2.49.

Comparing the individual years in our dataset, the distribution reacts relatively robustly to changes in the number of sources. The first year in our sample, August 2014 to July 2015, is once again the single outlier with a higher exponent of about 4, which could be the result of the much smaller number of SIPs in this year. While there were only 454 studies with a score of at least 50 in this period, we found more than a thousand in every other year. Although the number of SIPs differs in the years from August 2015 to July 2018 and lies between 1,193 SIPs and 2,213 SIPs (Fig 3A), the curve drops by similar exponents in all years (Table 4). The collective actions of individual science journalists lead to a long-tail distribution that approximates to a straight line on logarithmic scales in every year. For all years but 2015 to 2016, the goodness-of-fit tests show that the power law is not a plausible fit (Table 4). For most years, the exponentially truncated power law is supported as the best fit by likelihood ratio tests.

In summary, we could not confirm the hypothesis that a "pure" power law distribution is a plausible fit for the distribution of individual selections of scientific results. The comparison with other heavy-tailed distributions showed much stronger support for an exponentially truncated power law which is characterised by a steeper decline in the tail of the distribution.

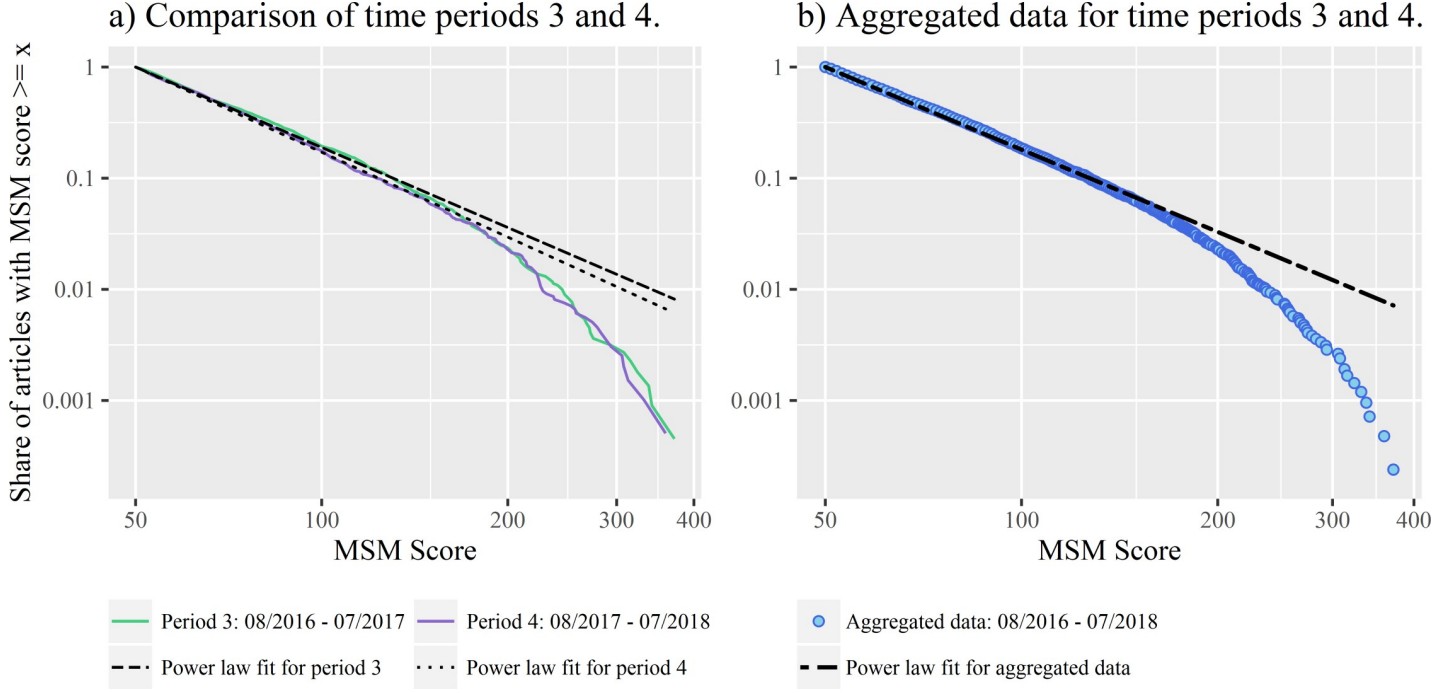

**Fig 5. Complementary Cumulative Distribution Functions (CCDFs) of the number of studies with a specific MSM score between August 2016 and July 2018**
**(N = 4,186).** A) CCDFs and power law fits for the share of studies with a certain MSM score for the last two years of the study period (N = 2,213 for the number of SIPs from August 2016 to July 2017 and N = 1,973 for August 2017 to July 2018). While the power law fit is a good representation of the data up to values of around 150, there is a steep drop-off in the tail. B) CCDF and power law fit for aggregated data from panel A).

## Discussion

The present study was, first of all, quite fundamentally about describing external effects [1]. We tried to identify a social structure resulting from the aggregated actions of individual science journalists, i.e. a collective of individual actors loosely linked by mutual observations who are concerned with the selection and preparation of scientific studies for public communication and who operate under similar restrictions. We tested the hypotheses that both the selection of scientific sources and the selection of concrete reporting events follow a power law distribution of the form $C^*x^{-\alpha}$. In both cases, we found that other heavy-tailed distributions are better fits for the data, especially the exponentially truncated power law. While the data shows a linear decline on logarithmic scales for a certain range, there is a steeper drop-off in the tail for both distributions and thus a deviation from the power law form. This could be

**Table 4. Parameters of power law and truncated power law fits and results of goodness-of-fit tests for the distribution of MSM scores in the different time periods.**

| Data | Power law: α (SE) | p-value | Exp. trunc. power law: α (SE) | Exp. trunc. power law: λ (SE) |
|---|---|---|---|---|
| Number of studies with a specific score, last two years aggregated | 3.45 (0.15) | < 0.0001 | 2.49 (0.16) | 0.0085 (0.0012) |
| Number of studies with a specific score, 2014–2015 | 4.03 (0.22) | 0.079 | 3.91 (0.30) | 0.0011 (0.0017) |
| Number of studies with a specific score, 2015–2016 | 3.31 (0.34) | 0.11* | 1.82 (0.62) | 0.013 (0.0032) |
| Number of studies with a specific score, 2016–2017 | 3.39 (0.19) | < 0.0001 | 2.32 (0.22) | 0.0092 (0.0016) |
| Number of studies with a specific score, 2017–2018 | 3.53 (0.17) | 0.0036 | 2.72 (0.24) | 0.0072 (0.0017) |

Standard errors of the parameters were estimated by bootstrapping.

*For p-values above 0.1, we do not reject the power law hypothesis.

explained by the fact that the number of studies published each year as well as the number of media outlets reporting on the studies are finite, leading to an upper bound for the possible MSM score and the possible number of studies with an MSM score $>= 50$ in a journal. Because of this upper bound, the distribution has to show a cut off in the tail [72].

We have described in more detail a social structure that has already been identified by other authors, mostly without them noticing it. It therefore seems reasonable to assume that this is a regularity that applies where individual journalists, guided by similar motives, have to choose from a supply of potential coverage events that they cannot overlook. This assumption can additionally be substantiated by the fact that a whole series of similar selection decisions follow similar heavy-tailed distributions, namely, for example, the number of citations of scientific articles, the number of hits on websites, or the number of books sold in the USA [91]. They are all social structures that result from selection decisions by individual actors who have to choose from an unmanageable number of options. One other study has also observed the existence of a power law distribution for topical pluralism in online journalism [92].

In the case of science journalism, a truncated power law can occur when we assume that the distribution of a new Social Impact Paper (SIP) among the available journals is proportional to the number of SIPs that journals have already had. If n new SIPs appear every year and if the distribution of these new SIPs is proportional to the quantity that journals have already had, a power law distribution will result (for mathematical proof see [72]). The distribution of journalistic attention to individual study results can be explained analogously. The attention gained for a certain type of study result will be distributed in proportion to the attention that this type achieved before. This lawfulness is generally referred to as the Matthew Effect or Yule Process (for overviews see [72,91]).

Sociologically, however, this explanatory approach is not fully convincing because it does not sufficiently include the actors' micro-perspective and the possibility of social change. In general, from a sociological point of view, these structures should be interpreted as a result of the interdependencies in the factual, temporal, and social dimensions outlined above. The relevance of the findings results from the assumption that the structure itself, i.e. the specific distribution of journalistic attention for scientific journals and individual study results, generally always emerges under the conditions of information overload. However, the specificity of the distribution, predominantly the exponent, but also the rank of journals or findings, is variable and subject to social change. We hope that our findings will help to better describe this change scientifically. Significant changes in one or more of the dimensions of influence are likely to affect the exponent of the distribution, not the distribution type itself. The exponent, together with the ranking, should provide information on the influence structure outlined and thus enable the analysis of social change in a quantifiable way.

However, it should be stressed that at this point, this indicator is nothing more than a plausible presumption. The main reason for this lies in the methodological approach of this study itself. It is difficult to assess the validity of the Altmetric MSM-score as an indicator for the journalistic selection for three reasons. Firstly, because it is based on the scraping of what we must assume to be a completely unsystematic selection of media titles; secondly, because many of these sources such as EurekAlert are unsuitable for indicating journalistic selection; and thirdly, because we find a large variance in the number of SIPs over the years of our study period. In order to be able to explore the suitability of the exponent of a distribution type more precisely, additional studies are necessary that reproduce the distribution pattern on a more valid data basis.

## Supporting information

**S1 File. Dataset of Social Impact Papers from August 2014 to July 2018.** Column "Dataset" contains codes for the individual years, 1 = August 2014 to July 2015; 2 = August 2015 to July 2016; 3 = August 2016 to July 2017 and 4 = August 2017 to July 2018.
(CSV)

## Acknowledgments

We wish to thank Lars Koppers for critically reviewing the manuscript and for advice concerning the data analysis and interpretation.

## Author Contributions

**Conceptualization:** Markus Lehmkuhl.

**Data curation:** Nikolai Promies.

**Formal analysis:** Nikolai Promies.

**Funding acquisition:** Markus Lehmkuhl.

**Investigation:** Markus Lehmkuhl, Nikolai Promies.

**Methodology:** Markus Lehmkuhl, Nikolai Promies.

**Project administration:** Markus Lehmkuhl.

**Resources:** Nikolai Promies.

**Software:** Nikolai Promies.

**Supervision:** Markus Lehmkuhl.

**Visualization:** Nikolai Promies.

**Writing – original draft:** Markus Lehmkuhl, Nikolai Promies.

**Writing – review & editing:** Markus Lehmkuhl, Nikolai Promies.

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
