## [Editor Report · Decision Letter 0]

28 May 2020

PONE-D-20-06379

Frequency distribution of journalistic attention for scientific studies and scientific sources: An input – output analysis

PLOS ONE

Dear Dr. Lehmkuhl,

Thank you for submitting your manuscript to PLOS ONE. After careful consideration, we feel that it has merit but does not fully meet PLOS ONE’s publication criteria as it currently stands. Therefore, we invite you to submit a revised version of the manuscript that addresses the points raised during the review process.

I was unsuccessful in finding reviewers for the manuscript so I reviewed it myself. My comments are attached.

We look forward to receiving your revised manuscript.

Kind regards,

Luís A. Nunes Amaral, Ph.D.

Academic Editor

PLOS ONE

Additional Editor Comments:

This review follows the Universal Principle Review approach. Lower numerical scores imply greater quality just as in NIH-style scoring of grant applications.

Journal Requirements:

2. You indicated that you followed methods given in ref. 84: "Kube L. Konsensanalyse der Berichterstattung in den Wissenschaftsressorts deutscher Tageszeitungen. Unveröffentlichte Bachelorarbeit. 2018." If materials, methods, and protocols are well established, authors may cite articles where those protocols are described in detail, but the submission should include sufficient information to be understood independent of these references (https://journals.plos.org/plosone/s/submission-guidelines#loc-materials-and-methods).

Summary:

This is an interesting well-written, if wordy, article on a very interesting topic that will be of interest to many researchers. The analyses reported are of good quality, but I have some recommendations that I believe will strengthen the manuscript.

Quality of research: Score 1.5 (Range 1-3)

The authors look at the statistical characteristics of the scientific articles receiving significant press attention both from the perspective of the inputs (in which scientific journals were these articles published) and the outputs (by which media were they covered).

The authors provide a very thorough review of the state of the art about this question including going to the point of analyzing some of the data reported in prior studies as a justification for formulating the two hypotheses they test in the study. They also provide a careful description of their approach to selecting data for analysis, and describe in detail their procedures for analyzing the data. Finally, they share some of their filtered data.

Next, I list my concerns and questions in the order of their appearance in the manuscript.

1. Table 1 includes the New Scientist as one of the sources for published research. However the New Scientist does not publish original research. I believe a note should be added to table caption letting the readers know that the authors are aware of this point.

2. The authors repeatedly write that "y decreases exponentially with x" (lines 224, 291, 303, 308, 320, 491, 503). This statement is incorrect. The correct statement is "log y decreases linearly with log x" or "y decreases exponentially with log x".

3. Since the hypothesis tested in the study involve power-laws, all graphs should be double-logarithmic. This applies already to Figs 1 and 2. For figure 2a, I recommend having the squares filled white since it will make it easier to distinguish the symbols in the figure.

4. I am not sure that the evidence from Fig. 1 is enough to propose that the exponent should be expected to be close to 1 since the data available is not a good sample of the entire population. This caveat is actually the reason why this study is important.

Concerning the power law exponent, its value is related to the concentration of impact. With regard to income and wealth, one number that quantifies their concentration in the hands of a few people is the Gini coefficient. A power law distribution with an exponent alpha > 1 results in a Gini coefficient of 1/(2*alpha - 1).

alpha = 1 yields G approaching 1, alpha = 2 yields G = 0.33

For comparison, a highly unequal society such as Brazil has G = 0.5 whereas for the EU G = 0.3.

5. In the discussion of the results in Figs 1 and 2, I would remove the R^2 values. I do not think the data is of good enough quality to warrant the quantification involved in the R^2 calculation.

6. In the Materials and Methods, the authors refer to the sources of the input and output data. I believe that Scopus is a very good source for publications. I think that Altmetric is not as good a source. The reason is that while coverage in Scopus has likely remained pretty constant during the study period, I doubt that the same is true for Altmetric. That means that results for 2014 are not comparable to results for 2018.

In fact, I strongly disagree with the statement in lines 333-335. The lack of coverage by Altmetric should left censor a large fraction of SIPs published in the earlier years of the studied period.

I suggest that the authors add a new figure showing (a) how the number of SIPs changes by time period, (b) how the number of distinct scientific journals publishing SIPs changes by time period, and (c) how the number of distinct news sources covers by Altmetric changes by time period.

7. I recommend removing Figs. 3 and 4 and replacing them with a 2 panel figure in which the first panel shows the data (use lines of different colors instead of circles) and a fit line for just the last time periods. The second panel shows the aggregated data for the years for which Altmetric sampling is appropriate (probably the last two years of the data).

I recommend that the same be done with Figs. 5 and 6.

8. I think that the discussion of model selection should come before showing quality of fits for the power law model and the values of the fit parameters.

Table 3 should appear before Table 2 and the Table should also include a comparison between the Weibull and the truncated power law.

Table 2 should be separate into two tables. And fit parameters should be given for both power law and truncated power law. By the way, the truncated poor law model seems to be the better one for both data sets.

Reproducibility of research: Score 1 (Range 1-3)

I believe that enough details are given for one to be able to reproduce most of the analyses reported.

However, it is not clear that the data retrieved from Altmetric for news sources is being shared (line 363). If it is not, it should be.

Completeness of research: Score 1.5 (Range 1-3)

The types of analyses included are perfectly adequate. However, as I mention above, I think that several of the analyses need to be performed in a more careful manner in order to strengthen the confidence on the results and conclusions.

It would be interesting to provide an analysis of the distribution of the topics of the SIPs. I realize that classifying nearly 6000 papers by hand is tricky. However, the authors could consider using the topic classification algorithm by Lancichinetti et al (Physical Review X 2015) on the titles and abstracts of the SIPs.

Or that analysis could be left for another paper...

Scholarship: Score 1 (Range 1-3)

I believe the authors do a really thorough job of reviewing the relevant literature.

Impact of research: Score 1.5 (Range 1-4)

I believe that the authors do a great job of motivating their study the results are quite interesting and strong in spite of all the caveats discussed in the manuscript.

Minor points:

1. In line 79, the authors write that "journalists can no longer unconditionally trust...". I am not sure the situation has really changed. what has maybe changes is the awareness of what is the potential for hyperbole or outright lies by scientists.

2. The sentence in line 171 "thus the bond between ..." is not at all clear to me.

3. In line 185, I think it should be "value" instead of "values"

4. Line 350, "for" instead of "from"

5. Line 414: Writing "fewer than 1 in 10,000" is more informative than "about 0.07 percent"
---

## [Author Response · Author response to Decision Letter 0]

14 Jul 2020

• You indicated that you followed methods given in ref. 84: "Kube L. Konsensanalyse der Berichterstattung in den Wissenschaftsressorts deutscher Tageszeitungen. Unveröffentlichte Bachelorarbeit. 2018." […]

o This was a reference error in the manuscript, we did not use methods from ref. 84, but from ref. 86 (“Clauset A, Shalizi CR, Newman MEJ. Power-Law Distributions in Empirical Data. SIAM Rev. 2009; 51: 661–703. doi: 10.1137/070710111.”) and we described these methods in detail in the Materials and methods part of our study. We have corrected the reference.

• Table 1 includes the New Scientist as one of the sources for published research. However the New Sci-entist does not publish original research. I believe a note should be added to table caption letting the readers know that the authors are aware of this point.

o We have added a footnote to Table 1 with the following content: “While the New Scientist was identi-fied as one of the most important external sources for the journalistic coverage of biomedical re-search in [67], it differs from the other sources listed in the table in that it does not publish original research. This also explains why the New Scientist cannot be found in the results of our analyses.”

• The authors repeatedly write that "y decreases exponentially with x" (lines 224, 291, 303, 308, 320, 491, 503). This statement is incorrect. The correct statement is "log y decreases linearly with log x" or "y de-creases exponentially with log x". 

o We have corrected all mentioned statements, writing that y decreases exponentially with the loga-rithm of x or that y decreases by 1/xα.

• Since the hypothesis tested in the study involve power-laws, all graphs should be double-logarithmic. This applies already to Figs 1 and 2. For figure 2a, I recommend having the squares filled white since it will make it easier to distinguish the symbols in the figure.

o We have changed Figure 1 and Figure 2 to logarithmic axes and changed the fillings of the symbols in Figure 2a. Additionally, we noticed that the references in the labels were wrong and have corrected them.

• I am not sure that the evidence from Fig. 1 is enough to propose that the exponent should be expected to be close to 1 since the data available is not a good sample of the entire population. This caveat is actual-ly the reason why this study is important.

o We have removed the value of the exponent from hypothesis 1.

• Concerning the power law exponent, its value is related to the concentration of impact. With regard to income and wealth, one number that quantifies their concentration in the hands of a few people is the Gini coefficient. A power law distribution with an exponent alpha > 1 results in a Gini coefficient of 1/(2*alpha - 1).

o We think that a discussion of the relationship between the Gini coefficient and the power law expo-nent would exceed the scope of our study. But we thank the editor for this comment, we are already working on another study in which we focus on the Gini coefficient of the distributions of source se-lection and selection of single results in science journalism. In this study, we will calculate the Gini coefficient independent of a power law fit.

• In the discussion of the results in Figs 1 and 2, I would remove the R^2 values. I do not think the data is of good enough quality to warrant the quantification involved in the R^2 calculation. 

o We have removed the R^2 values.

• In the Materials and Methods, the authors refer to the sources of the input and output data. I believe that Scopus is a very good source for publications. I think that Altmetric is not as good a source. The reason is that while coverage in Scopus has likely remained pretty constant during the study period, I doubt that the same is true for Altmetric. That means that results for 2014 are not comparable to results for 2018.

In fact, I strongly disagree with the statement in lines 333-335. The lack of coverage by Altmetric should left censor a large fraction of SIPs published in the earlier years of the studied period.

I suggest that the authors add a new figure showing (a) how the number of SIPs changes by time period, (b) how the number of distinct scientific journals publishing SIPs changes by time period, and (c) how the number of distinct news sources covers by Altmetric changes by time period.

o We have changed the paragraph mentioned above (lines 333-335) and changed its position in the text. We have added a figure showing a) and b) (Fig 3a,b), but we are not able to show c) because Altmetric does not publish information concerning the development of the source base they use to calculate the MSM score. Additionally, we discuss the implications of the changes in the number of SIPs and journals with SIPs for our results in the results and discussion part and only use the last two years for the aggregated fits.

• I recommend removing Figs. 3 and 4 and replacing them with a 2 panel figure in which the first panel shows the data (use lines of different colors instead of circles) and a fit line for just the last time periods. The second panel shows the aggregated data for the years for which Altmetric sampling is appropriate (probably the last two years of the data).

I recommend that the same be done with Figs. 5 and 6

o We have created two new figures (Fig 4a, b and 5a, b) showing only the data and fits for the last two years.

• I think that the discussion of model selection should come before showing quality of fits for the power law model and the values of the fit parameters.

o We feel that changing the order of these paragraphs would not correspond to the logic of our analy-sis. Our overall goal was to test the hypotheses that a power law is a plausible model for the journal-istic selection of scientific sources and of single study results. Our focus was mostly on the power law distribution. Therefore, our first steps were to estimate the power law fits and check their good-ness-of-fit. Only after these steps, we could compare our model to other candidate distributions. While we understand the editor’s recommendation, we would like to leave the order of these para-graphs as it is. We have added an additional paragraph in the discussion in which we stress that other distributions are better fits.

• Table 3 should appear before Table 2 and the Table should also include a comparison between the Weibull and the truncated power law.

Table 2 should be separate into two tables. And fit parameters should be given for both power law and truncated power law. By the way, the truncated poor law model seems to be the better one for both data sets.

o We have changed the order of tables and added a column for the missing comparison. We have separated table 2 into two tables and added columns for the alpha- and lambda-parameter of the truncated power law.

• However, it is not clear that the data retrieved from Altmetric for news sources is being shared (line 363). If it is not, it should be.

o We do share a dataset containing all SIPs with their title, authors, the journals they were published in, the DOI, the year of publication and the MSM score (S1 File). We are not able to share data about the individual news sources that mentioned the studies because we did not have access to this in-formation. We only retrieved the value of the MSM score.

• It would be interesting to provide an analysis of the distribution of the topics of the SIPs. I realize that classifying nearly 6000 papers by hand is tricky. However, the authors could consider using the topic classification algorithm by Lancichinetti et al (Physical Review X 2015) on the titles and abstracts of the SIPs.

Or that analysis could be left for another paper...

o We think that an analysis of the distribution of the topics of the Social Impact Papers exceeds the scope of this paper, in which our main goal is to determine the structure of the journalistic selection of research results. But we agree that it will be interesting to conduct such an analysis and would like to thank the editor for recommending an algorithm that could be used to classify the texts automatically.

• In line 79, the authors write that "journalists can no longer unconditionally trust...". I am not sure the situation has really changed. what has maybe changes is the awareness of what is the potential for hyper-bole or outright lies by scientists.

o We think that it is not important whether the situation or journalists’ awareness of the situation has changed, because the result would be the same. That being said, we would argue that both the sit-uation and the awareness for the situation have changed.

• The sentence in line 171 "thus the bond between ..." is not at all clear to me.

o We have changed the sentence to “Such numbers illustrate the extent to which the bond between journalism and a sufficiently large audience is eroding.”

• In line 185, I think it should be "value" instead of "values"; Line 350, "for" instead of "from"

o We have changed both words.

• Line 414: Writing "fewer than 1 in 10,000" is more informative than "about 0.07 percent"

o We have changed the wording, but it is “fewer than 1 in 1,000”, not “fewer than 1 in 10,000”.

---

## [Editor Report · Decision Letter 1]

6 Aug 2020

PONE-D-20-06379R1

Frequency distribution of journalistic attention for scientific studies and scientific sources: An input – output analysis

PLOS ONE

Dear Dr. Lehmkuhl,

Thank you for submitting your manuscript to PLOS ONE. After careful consideration, we feel that it has merit but does not fully meet PLOS ONE’s publication criteria as it currently stands. Therefore, we invite you to submit a revised version of the manuscript that addresses the points raised during the review process.

We look forward to receiving your revised manuscript.

Kind regards,

Luís A. Nunes Amaral, Ph.D.

Academic Editor

PLOS ONE

Additional Editor Comments (if provided):

Thank you for all the revisions to the manuscript. I believe the manuscript is now nearly ready for publication. I just have a few of minor stylistics issues:

1. Please always include the leading zero in numerical values '0.0001' instead of '.0001' (see, example, Tables 2-4)

2. In Tables, you list values of the estimate and standard deviation. Is it really the standard deviation (SD) or is it the standard error (SE)? The SE of the estimate of a coefficient is more useful than the SD.

2. When presenting estimates for coefficients, do not include non-significant digits. For example, '2.4868 (0.1579)' should be replaced with '2.48 (0.16)' or '2.5 (0.2)'

---

## [Author Response · Author response to Decision Letter 1]

18 Aug 2020

• Please always include the leading zero in numerical values '0.0001' instead of '.0001' (see, example, Tables 2-4). When presenting estimates for coefficients, do not include non-significant digits. For example, '2.4868 (0.1579)' should be replaced with '2.48 (0.16)' or '2.5 (0.2)'

 o We have added leading zeros where missing and removed non-significant digits in all tables and in the text.

• In Tables, you list values of the estimate and standard deviation. Is it really the standard deviation (SD) or is it the standard error (SE)? The SE of the estimate of a coefficient is more useful than the SD.

 o The values listed in the tables are the standard deviation of the parameters in the bootstrap samples, i.e. the bootstrapped standard error. We have changed SD to SE and added a short explanation under tables 3 and 4 (“Standard errors of

 the parameters were estimated by bootstrapping.”).

---

## [Editor Report · Decision Letter 2]

3 Sep 2020

PONE-D-20-06379R2

Frequency distribution of journalistic attention for scientific studies and scientific sources: An input – output analysis

PLOS ONE

Dear Dr. Lehmkuhl,

Please note that in your last revision, rounding of decimal numbers was incorrectly done. 

E.g. when rounding a value like 0.9898 (as in page 22) to one significant digit, you have to round to the closest digit, in this case 1.0 and not 0.9. Please look at the section rounding decimals here: https://www.mathsisfun.com/rounding-numbers.html

Please, submit a revised version of the manuscript following this advice.

Submit your revised manuscript by Oct 18 2020 11:59PM. If you will need more time than this to complete your revisions, please reply to this message or contact the journal office at plosone@plos.org. Please include the following items when submitting your revised manuscript:

We look forward to receiving your revised manuscript.

Kind regards,

Miguel A Andrade-Navarro, Ph.D.

Academic Editor

PLOS ONE

---

## [Author Response · Author response to Decision Letter 2]

9 Sep 2020

Dear Editor,

we have again checked each decimal number and rounded all of them to two decimals or at least two significant digits. We hope that they are correct now. The example of an incorrect rounding you gave in your review (0.9898 to 0.9 on page 22) is probably based on a misreading caused by the tracked changes mode. The number we rounded to 0.9 was 0.898 and not 0.9898. We have now changed it to 0.90. The version of the manuscript with tracked changes we uploaded shows the changes from the unrounded numbers as they were before the revision you criticized.

---

## [Editor Report · Decision Letter 3]

14 Oct 2020

Frequency distribution of journalistic attention for scientific studies and scientific sources: An input – output analysis

PONE-D-20-06379R3

Dear Dr. Lehmkuhl,

We’re pleased to inform you that your manuscript has been judged scientifically suitable for publication and will be formally accepted for publication once it meets all outstanding technical requirements.

Kind regards,

Miguel A Andrade-Navarro, Ph.D.

Academic Editor

PLOS ONE
---

## [Editor Report · Acceptance letter]

20 Oct 2020

PONE-D-20-06379R3 

Frequency distribution of journalistic attention for scientific studies and scientific sources: An input – output analysis 

Dear Dr. Lehmkuhl:

I'm pleased to inform you that your manuscript has been deemed suitable for publication in PLOS ONE. Congratulations! Your manuscript is now with our production department. 

Kind regards, 

on behalf of

Dr. Miguel A Andrade-Navarro 

Academic Editor

PLOS ONE